# Liquefied Synthetic Natural Gas Produced through Renewable Energy Surplus: Impact Analysis on Vehicular Transportation by 2040 in Italy

**Linda Barelli \*** , **Gianni Bidini, Panfilo Andrea Ottaviano** and **Michele Perla**

Department of Engineering, University of Perugia, via G. Duranti 1/4A, 06125 Perugia, Italy;
gianni.bidini@unipg.it (G.B.); panfilo.ottaviano@unipg.it (P.A.O.); mic.perla@gmail.com (M.P.)
\* Correspondence: linda.barelli@unipg.it; Tel.: +39-075-5853740

**Abstract:** Time mismatch between renewable energy production and consumption, grid congestion issues, and consequent production curtailment lead to the need for energy storage systems to allow for a greater renewable energy sources share in future energy scenarios. A power-to-liquefied synthetic natural gas system can be used to convert renewable energy surplus into fuel for heavy duty vehicles, coupling the electric and transportation sectors. The investigated system originates from power-to-gas technology, based on water electrolysis and $CO_2$ methanation to produce a methane rich mixture containing $H_2$, coupled with a low temperature gas upgrading section to meet the liquefied natural gas requirements. The process uses direct air $CO_2$ capture to feed the methanation section; mol sieve dehydration and cryogenic distillation are implemented to produce a liquefied natural gas quality mixture. The utilization of this fuel in heavy duty vehicles can reduce greenhouse gases emissions if compared with diesel and natural gas, supporting the growth of renewable fuel consumption in an existing market. Here, the application of power-to-liquefied synthetic natural gas systems is investigated at a national level for Italy by 2040, assessing the number of plants to be installed in order to convert the curtailed energy, synthetic fuel production, and consequent avoided greenhouse gases emissions through well-to-wheel analysis. Finally, plant investment cost is preliminarily investigated.

**Keywords:** synthetic LNG; renewable fuels; well-to-wheel; vehicular transportation





## 1. Introduction

The increase in renewable energy source (RES) penetration in the energy scenario, as remarked in the Paris Climate Conference COP21 [1], is a fundamental element for the future clean energy transition. To this aim, energy storage systems (ESSs) integration is necessary to avoid risks for the operational stability of the power system [2]. The cooperation for the strategic development of an energy system in the EU is defined by Regulation (EC) No 714/2009 of the European Parliament and of the Council; consequently, community-wide ten-year network development plans are published for electricity and gas transmission system operators (TSO). National energy development has to take into account the reference scenarios shown in community plans; specifically for Italy, the following three different 2040 scenarios are described in the literature [3]: business-as-usual (BAU), based on the present trend; centralized (CEN); and decentralized (DEC). The second and third scenarios are compliant with the decarbonization target, final energy consumption, and RES penetration for 2030 [4]. CEN is based on consumption reduction and programmable RES development with the existing gas infrastructure. In DEC, an increase of non-programmable RES and a higher electrical consumption share are expected. The DEC scenario shows the needs of ESS development and grid flexibility improvement in order to contain the intrinsic overgeneration. Consequently, power-to-gas technology is a key for sector coupling, producing hydrogen or syngas to be injected into the gas grid [5], as investigated elsewhere [6,7] for several case studies. Moreover, the expected ESS growth

in the 2040 DEC scenario, for pumped hydro and electrochemical storage technologies [3], is set to 11.5 GW with respect to 2017.

Several energy storage strategies are described in the literature—the production of fuel via electrolysis is characterized by a high capacity and long term-storage, supporting different demand sectors such as thermal, transport, and industrial uses, as well as power generation. In a power-to-X concept, liquefied synthetic natural gas (LSNG) constitutes an energy carrier that shows advantages in volumetric energy density (5.8 kWh/L) when compared with compressed natural gas (2.5 kWh/L at 250 bar), compressed hydrogen (1.4 kWh/L at 700 bar), or liquefied hydrogen (2.3 kWh/L) [8]. The gas mixture produced via electrolysis and methanation must be processed in order to remove water and $CO_2$ so as to avoid solid phase formation during liquefaction; moreover, the $H_2$ content must be reduced for engine utilization or grid injection after regasification. A P2LNG process could have a significant impact on the greenhouse gas (GHG) reduction for heavy duty vehicles (HDV) in the transportation sector. Nowadays, there are over 2000 trucks fed by liquefied natural gas (LNG) in Italy and fuel distribution is available in over 80 stations across the country [9], with an expected import of 1,250,000 tonnes/year in 2025 [10]. When LNG is used as truck fuel, the corresponding GHG emissions per kilometer are reduced by up to 20% with respect to diesel. In the case of renewable LSNG utilization, a reduction of up to 92% could be reached, with negligible SOx emissions [11]. A power-to-liquefied synthetic natural gas (P2LSNG) system can be used to provide energy storage [2] with the consequent mitigation of RES curtailment, and, at the same time, can produce renewable fuel to improve the renewable share in transportation [12]. LSNG can also be used after regasification in ordinary compressed natural gas (CNG) fueled vehicles and even in different sectors, such as electricity generation or grid injection to supply distribution to industrial or civil users when the downstream systems can tolerate the $H_2$ content (a specific evaluation is necessary for domestic utilization).

## 2. Methodology

The DEC scenario is characterized by the growth of the electrical share in the final energy consumption with the diffusion of heat pumps and electrical vehicles, and the phase out of coal fired thermoelectrical power plants by 2025 and a strong growth of distributed RES plants. Commercial vehicle quantities are considered equal between diesel, electrical, and CNG/$H_2$ technologies; moreover, electrical conversion is not developed for HDV and natural gas consumption for transportation is expected to grow. A P2LSNG process, presented by authors in a previous work [13], is applied here to evaluate the impact on vehicular transportation using the RES curtailed energy in Italy to obtain the results for the 2040 forecasts under the DEC scenario. In previous work [13], an impact analysis was already performed on the Irpinia territory in southern Italy. In this area, the curtailed energy amount and transmission lines criticality were known; the $CO_2$ sources to feed the methanation section were consequently evaluated and identified in the same territory (as emission trading system or biogas plants). The utilization of the obtainable LSNG production was assessed at about 23.7 Mkm/y in HDV transport (corresponding to 8.5% of HDV traffic on the local A16 highway), and 55,652 tonnes$CO_2$e/y of emissions were avoided, as determined through a well-to-wheel (WtW) comparison with diesel powered HDV.

For the national analysis presented here, the P2LSNG process was integrated with a $CO_2$ direct air capture (DAC) section. This option reduces the power-to-chemical efficiency of the process, as the auxiliary loads (electric and thermal) increase. The DAC technology allows one to considering the $CO_2$ required to feed the methanation, as available at the RES curtailed plants, independently from the geographical location of the local $CO_2$ sources, thus avoiding the need for a pipeline. Specifically, high temperature chemical absorption technology is chosen for DAC; all of the specifications are detailed in Section 3.1. Alternatively, as a potential $CO_2$ source, the upgrade of biogas produced at a national level was also considered. In order to calculate the optimal number of plants to exploit the whole national curtailed energy, a load duration curve was built over the 2040 gross overgenera-

tion data (Figure 1). These data were calculated on the basis of the RES production and electric demand projections as reported in the DEC scenario published by Terna SpA and Snam Rete Italia SpA in 2019 [3]. The curve represents a potential gross overgeneration, as it does not take into account for short-term grid storage requirements and for the impact of additional storage systems with respect to the existing ones.

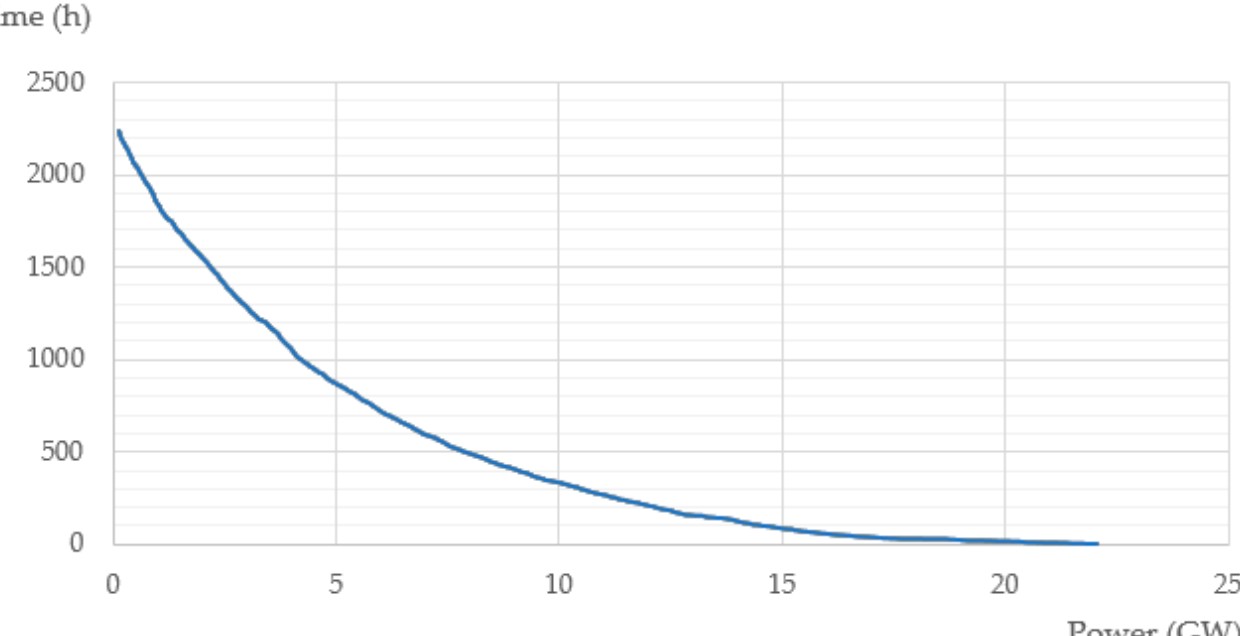

**Figure 1.** Duration curve of decentralized (DEC) scenario overgeneration in 2040.

The minimum value for the plant utilization factor was set to 20%. Moreover, in the case of DAC integration, the installed power of each P2LSNG plant was evaluated considering the auxiliary electric consumption of the DAC section, as discussed in Section 3.1. Consequently, it was increased by up to 25.51 MWel with respect to the 21.5 MWel SOE installed power determined in the literature [13]. The number $N_Z$ of P2LSNG plants was determined by Equation (1).

$$N_Z = \frac{P_{20\%}}{P_{LSNG}} \qquad (1)$$

where $P\_(20\%)$ is the total power to be installed, determined as the power value of the overgeneration duration curve, corresponding to 20% time utilization over the year. It results in 1284 MWel.

$P\_LSNG$ is the SOE installed power, equal to 25.51 MWel.

According to Equation (1), the number of P2LSNG plants to be installed are rounded down to 50. Consequently, the maximum power $P\_max$ that can be used to produce LSNG results in 1275 MWel. This value, according to Figure 1, can be exploited for 2242 h per year. Therefore, the maximum energy utilization is calculated through the time integration of the load duration curve up to $P\_max$, as described in Equation (2).

$$E_{av} = 10^{-3} \cdot \int_0^{P_{max}} t dP \qquad (2)$$

The yearly curtailed energy $E\_av$ results in 2551 GWh/y, which can be exploited to feed the P2LNG plants.

The impact of plant operation is evaluated at national level on wind energy curtailment and transport emissions in Section 4; for this purpose, the WtW analysis is applied to calculate the GHGs avoided when the produced fuel utilization for HDV is compared with

diesel. The DEC scenario does not include a heavy transport demand for 2040; the freight transport activity is set at 328 Gtkm for 2030 in DEC and CEN, a similar value for 2030 (323 Gtkm) is provided in the literature [14], then increased up to 347 Gtkm in 2040. This value will be compared to 271 Gtkm for 2020 in order to estimate the growth of traffic levels. Consequently, the HDV traffic in the 2040 DEC scenario will be estimated.

### 3. LSNG Process

The P2LSNG is based on a solid oxide electrolyser (SOE) section fed by renewable energy. This technology is chosen for its high efficiency, which is significantly greater than for PEM technology. Moreover, the sizing made on the basis of the power value and the operation time deduced from the overgeneration duration curve allows for limiting the need for load regulation.

The SOE is coupled with a methanation section, which was developed elsewhere [15], and a DAC section, which provides the $CO_2$ feed to the methanation reactor, where a methane rich mixture is produced (composition in Table 1). A gas treatment section, as developed elsewhere [13], is used to remove impurities and to guarantee the required LNG quality mixture [16]. A temperature swing adsorption (TSA) dehydration section based on zeolite sorbents is used to reduce the water content in the outlet gas to below 0.1 ppmv. This threshold was chosen to avoid ice formation or solid hydrates, and the consequent risk of valves and equipment clogging. A cryogenic dual pressure distillation system allows for reducing the $CO_2$ concentration to below 50 ppmv in order to prevent dry ice formation, and the $H_2$ content below 2% to enable fuel exploitation in internal combustion engines (ICE) and natural gas tanks [17,18]. An additional solid oxide fuel cell (SOFC), fed by the hydrogen separated in the distillation section, covers the chillers' electrical demand and provides a part of the process thermal demand. Finally, the residual thermal demand was supplied by a burner fed by a portion of the produced LSNG. Figure 2 reports a plant functional scheme.

**Table 1.** Characteristics of the produced raw hydromethane.

| Case | Value |
| --- | --- |
| $X_{CH4}$ | 0.52 |
| $X_{CO2}$ | 0.38 |
| $X_{H2}$ | 0.084 |
| $X_{H2O}$ | 0.016 |

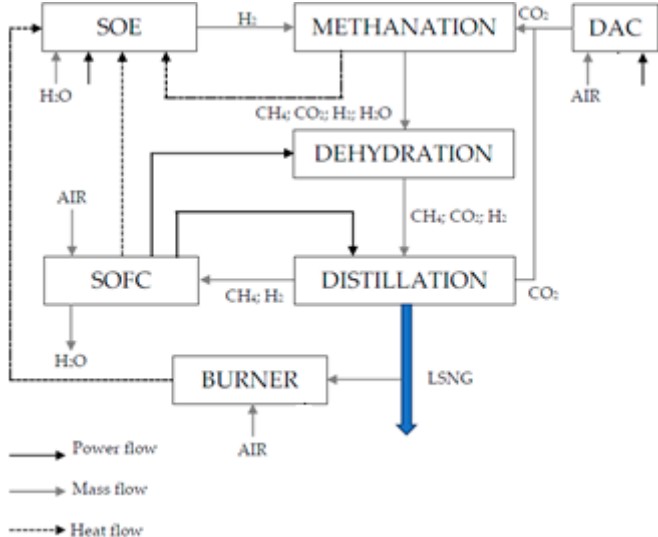

**Figure 2.** Proposed power-to-liquefied synthetic natural gas (P2LSNG) system scheme according to the literature.

### 3.1. Direct Air Capture System (DAC)

Different technologies can be used for DAC; here, high temperature chemical absorption was chosen. Specifically, ambient air is injected in a contactor column where sodium hydroxide (NaOH) reacts with $CO_2$, producing an aqueous solution of sodium carbonate ($Na_2CO_3$). This solution is sent to a regeneration column where it reacts with calcium hydroxide ($Ca(OH)_2$), producing calcium carbonate ($CaCO_3$) and NaOH. Subsequently, sodium hydroxide is recycled to the first column. $CaCO_3$ is then heated to release $CO_2$ and CaO, which reacts with water regenerating $Ca(OH)_2$. Calcium hydroxide is then sent back to the second column. The $CO_2$ separation is obtained at 900 °C, the heat demand is between 1420 and 2250 kWhth/t$CO_2$, and the electric demand for air and solutions transport is 366–764 kWhel/t$CO_2$ [19].

In the proposed P2LSNG process, the DAC electrical demand is provided by RES curtailed energy. In order to avoid fuel consumption penalties, the selected DAC system uses potassium hydroxide (KOH) as an absorber [20]. This system can be implemented to be fully electrified with a demand of 1535 kWhel/t$CO_2$. Moreover, it can be considered more efficient when compared with low temperature DAC technologies, as discussed in [19], which provides an overview on state of the art of DAC technologies. The $CO_2$ net demand at the methanator is 2.613 t/h, and the additional electric demand (with respect to the power required by the SOE) for the P2LSNG is then 4.01 MWel. The resulting electric demand for the whole process (P_LSNG in (1)) is 25.51 MW.

### 3.2. Electrolysis and Methanation

The electrolysis section of the process consists of a 21.5 MWel SOE [13], where hydrogen is produced according to the configuration detailed previously in the literature [15], with a high electrical efficiency of 76% (not considering heat for steam production). The SOE output stream is mixed with the $CO_2$ provided by the DAC section, thus the final produced LSNG can be considered to be a carbon neutral fuel. The methanation reaction takes place in a three-stage nickel catalytic reactor, converting the $H_2$ and $CO_2$ mixture into $CH_4$ and $H_2O$, with an efficiency of 83.4%. Heat recovery from the exothermic methanation is used to reduce the SOE thermal demand estimated at 3.97 MWth. The produced mixture, as described in Table 1, is characterized by a high methane content, and it must be processed to improve its chemical energy content (LHV is 18.19 MJ/Smc). Moreover, the raw mixture must be treated to remove $CO_2$ and $H_2O$ in order to avoid technical issues in the liquefaction process. In addition, a specific evaluation on the infrastructure materials has to be performed in the case of the methane and hydrogen mixture injection into the gas grid. Thus, limits for $H_2$ concentration are not specified in Italian regulations [21] according to the European standard EN 16726. The main criticality is represented by hydrogen permeation, leading to metal embrittlement, as well as risk of fracture and leak. The limit for hydrogen utilization in ICE is attested at 2%, as previously specified.

### 3.3. Dehydration and Distillation

As previously detailed, the produced methane rich mixture must be processed with the consumption of both electricity and heat. The P2LSNG system configuration [13] was optimized in order to simplify the process layout, reducing the energy demand. As reported in a review paper [22], temperature swing adsorption (TSA), out of all of the gas drying techniques, is the one that guarantees LNG quality water removal. In the proposed system, TSA is preferred over other technologies that need a more complex plant scheme and higher energy demand [23,24]. The dehydration section is composed of two batch reactors containing zeolite 3A sorbent [25], as described in Figure 3; specifically, adsorption is operated at 13 bar at an ambient temperature and desorption is operated at 280 °C, switching operation after 8 h. Heat recovery is obtained through two heat exchangers and the dehydrated gas is compressed at 48 bar. After dehydration, the gas upgrading is obtained in a distillation section composed of three columns. In the first two, $CO_2$ is separated using a dual pressure cryogenic distillation scheme, as proposed in the

literature [26], with a low energy demand when compared to other systems [27,28]. The high (HP) and low pressure (LP) columns work at 48 bar and 40 bar, respectively.

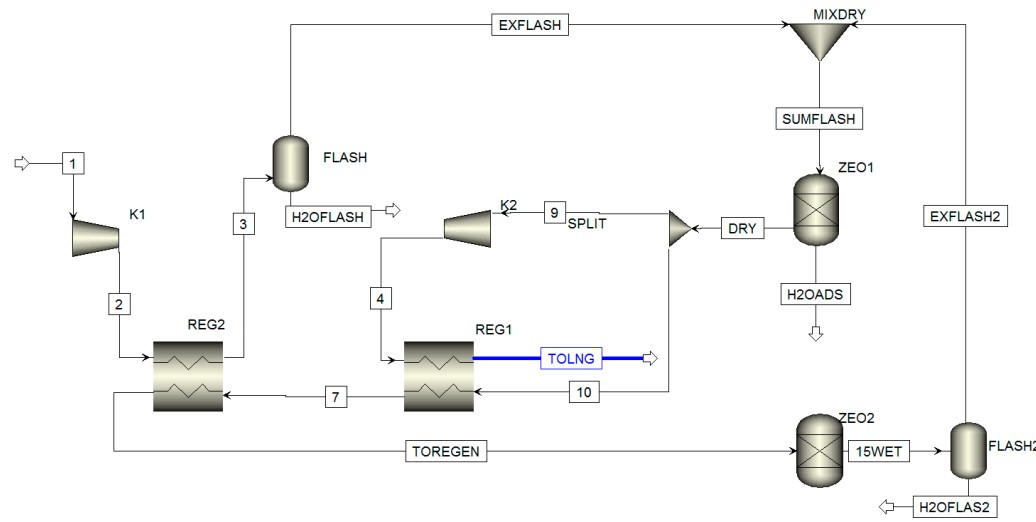

**Figure 3.** Dehydration section scheme [13].

The separated $CO_2$ (stream "2E" in Figure 3) is sent to the methanation section. The third distillation column ("$H_2COL$") is operated at 38 bar to separate the $H_2$. In particular, LSNG is the tail product (stream "$CH_4$", with a methane concentration of 98.18%) and a hydrogen–methane mixture leaves the top of the column ("$H_2$"). The latter is then expanded down to 3 bar and is sent to the SOFC section. Finally, four heat exchangers, as depicted in Figure 4, allow for heat and cold recovery.

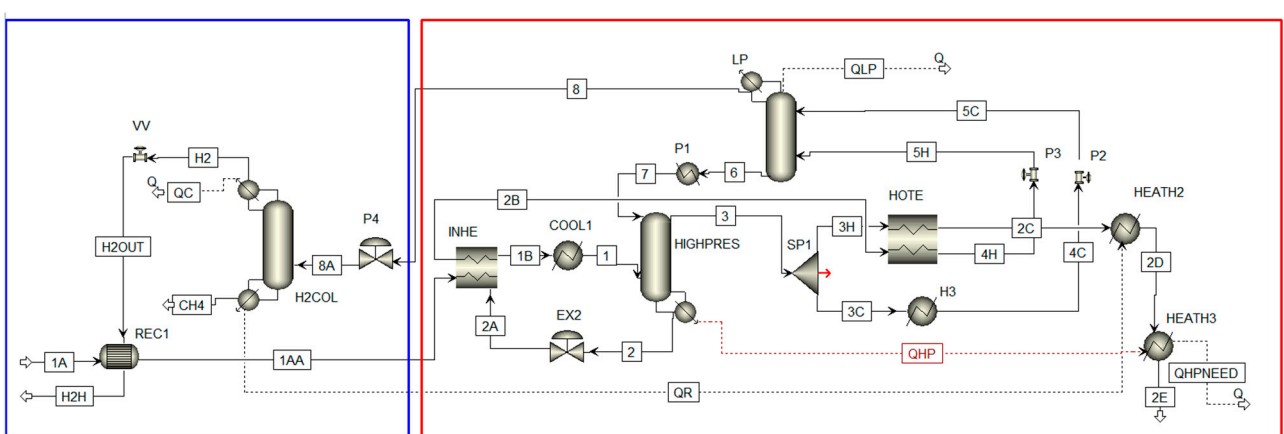

**Figure 4.** Low temperature distillation scheme [13].

### 3.4. SOFC Section for Energy Recovery

A SOFC section, modelled in the literature [29], is used to cover the internal electricity demand, exploiting the $H_2$-$CH_4$ mixture resulting from the cryogenic upgrade section ("$H_2$" stream of Figure 3). Specifically, the SOFC electrical power output is 1.5 MWel and the recoverable heat is about 1.1 MWth at 850 °C. The electric demand for compression and cooling duties reaches 1.05 MWel for the whole P2LSNG system, so the residual produced energy is used for auxiliary systems. With regards to the thermal balance, considering the heat required for vaporization, in the SOE section, and for reboilers, in the distillation section, the whole system heat demand reaches 4.1 MWth. It is clear, therefore, that internal heat recovery is not sufficient to cover this thermal load, hence 53 kg/h of LSNG, equal to

7% of the production, is burned to supply the residual heat demand. A detailed description of the working conditions, mass, and electric and thermal energy streams for all sections can be found elsewhere [13]. The PLSNG electrical to chemical efficiency is then calculated through Equation (3).

$$\eta = \frac{LSNG_{net,prod}(MW_{LHV})}{P_{LSNG}(MW_{el})} \tag{3}$$

where $LSNG_{net,prod}(kW_{LHV})$ is the produced chemical power equal to 10.689 $MW_{LHV}$ (LSNG output equal to 761 kg/h) and $P_{LSNG}$ considering the DAC consumption, is 25.51 $MW_{el}$; considering these values, the process efficiency results in 41.9%.

## 4. Impact Analysis

Based on the available curtailed energy in the DEC scenario for 2040, the authors investigate the implementation of the proposed P2LSNG system at a national level, through the realization of 50 plants. Specifically, the authors evaluate the GHG reduction, as a result of LSNG utilization as HDV fuel, applying the WtW method. The Italian electric system for these plants types in 2016 was characterized by RES overgeneration, mostly caused by the wind power capacity; curtailment operations, by the national TSO Terna S.p.A., reached 325 GWh for these plants types in 2016, increasing up to 436 GWh in 2017 [30]. As reported previously [27], the implementation of the proposed LSNG process was already investigated by the authors in a specific area of southern Italy, called Irpinia. It resulted in an LSNG production covering 8.5% of HDV traffic on the local A16 highway, avoiding the emissions of 55,652 tonnesCO$_2$e/y when compared with diesel powered HDV. Looking at the DEC 2040 scenario, an expected overgeneration of 11,214 GWh can be calculated on a yearly basis. According to Equation (2), about 2551 GWh of wasted energy could be used to produce LSNG, with a reduction on curtailment of 22.7%. Equaton (4) provides the calculation of the produced fuel.

$$LSNG = \frac{\eta \cdot E_{av} \cdot 3600}{LHV_{LSNG}} \tag{4}$$

where *LSNG(ton)* is the total amount of fuel produced by the proposed process (76,657 ton), $\eta$ is the power-to-chemical efficiency (41.86%), $E_{av}$ *(GWh$_{el}$)* is the available energy (2551.32 GWh), and $LHV_{LSNG}$ (MJ/kg) is equal to 50.16 MJ/kg.

Through the WtW analysis, emissions are calculated in gCO$_2$e/MJLSNG. This analysis is usually applied considering two parts of the lifecycle.

First, the well-to-tank (WtT) one, which considers fossil LNG [31] remote production, shipping to Europe, terminal operation, distribution to refueling stations, and refueling operations. In this work, gas production is supported by RES and the produced CO$_2$ after combustion is captured from the ambient air, making the process carbon neutral. The CO$_2$ emissions for distribution to refueling stations can be considered negligible because of a growing commercial network, which nowadays covers almost the whole national territory. A refueling boil-off component is assumed at 0.5 gCO$_2$e/MJLSNG as a base case value [31].

Second, the utilization in HDV completes the lifecycle with the tank-to-wheel (TtW) analysis. LNG conversion in ICE is evaluated in 173.8 gCO$_2$e/MJout for high pressure direct injection engines (HDPI) [31]. In the LSNG case, the CO$_2$e due to combustion is captured from the atmosphere; hence, this component can be considered equal to the bio-LNG case, evaluated at 26.1 gCO$_2$e/MJLSNG [32]. Table 2 reassumes the WtW comparison between LSNG and diesel utilization in HDV, considering 262 gCO$_2$e/MJout as the reference for diesel emissions in WtW reports [31].

**Table 2.** Well-to-wheel (WtW) analysis for liquefied synthetic natural gas (LSNG) compared with diesel.

| | GHG Emissions (gCO$_{2e}$/MJ$_{out}$) | |
|---|---|---|
| | **LSNG** | **DIESEL** |
| Gas Production | - | - |
| Liquefaction | - | - |
| Shipping and terminal | - | - |
| Distribution and refueling | 0.5 | - |
| WtT total | 0.5 | 47 |
| TtW | 26.1 | 215 |
| WtW | 26.6 | 262 |

In order to calculate the avoided emissions in CO$_2$etonnes/year, the average diesel and LSNG specific consumption and LHV values [33] are reported in Table 3.

**Table 3.** Emission calculation reference values. * (98.18% CH$_4$—1.81% H$_2$—CO$_2$ < 50 ppm).

| | **LSNG** | **DIESEL** |
|---|---|---|
| Consumption | 25 (kg/100 km) | 30 (L/100 km) |
| LHV | 50.16 (MJ/kg) * | 36 (MJ/L) |

The current HDV traffic on national highways is equal to 19,116 Mkm [34] (24,476 Mkm can be evaluated for 2040 DEC [35]), and its growth is expected, as indicated in Section 2. The produced LSNG, equal to 76,657 tonnes/y, can be used to cover 306.63 Mkm. According to Tables 2 and 3, the produced GHG resulting from total LSNG utilization amounts to 102,281 tonnesCO$_2$e/y; in the case of diesel utilization, this calculation results in about 867,643 tonnesCO$_2$e/y, with consequent avoided emissions of 765,362 tonnesCO$_2$e/y. The outlook for the LNG demand at 2030 is estimated as 0.8 Mtonnes/y for vehicle utilization [36], and assuming a conservative constant growth until 2040, a demand of 1.8 Mtonnes/y could be expected; hence, the P2LSNG plant lot production could cover 4.2% of the national demand by 2040.

## 5. Cost Evaluation

The capital investment cost is estimated for each section of the proposed P2LSNG plant, including the installation, balance of plant, and engineering. The method applied here according to Equation (5) is based on scale factors, known capital costs, train number, and size [36].

$$C = C_0 \cdot \left(\frac{S}{S_0}\right)^f \tag{5}$$

where $C$ is the capital cost in M€ for a plant section with size $S$, $C_0$ is the cost in reference for a plant section with size $S_0$, and f is the scale factor. A fixed Euro/Dollar exchange rate of 1.19 was taken as a reference. The values for each plant section are reported in Table 4.

The $C_0$ value for the DAC section is chosen considering the expected development maturity of this technology in a 2040 scenario [19]. The SOE section cost is evaluated from the automated production scenario described in [37] (around 308 €/kW); this value is similar to capital expenditure outlook for 2040 (300 €/kW [40]). A conservative scale factor for system downsizing is then applied to calculate the cost for the 21.5 MW section. The cost of the cryogenic section is evaluated as described elsewhere [41–43]. Related costs are considered to be equal to the ones of the biogas cryogenic upgrade process for bio-LNG production (as the two processes have similar characteristics), considering the minimum temperature and two column configuration. In the reported evaluation [41–43], the capital cost related to biogas distillation plant is assumed to be equivalent to the one of an air separation unit (ASU), because of the presence of two columns, namely, a minimum temperature level (−79 °C in biogas and −160 °C in ASU) and similar heat exchanger pinch point temperature. As the

proposed process configuration includes three distillation columns, with the last operating at a minimum temperature of $-115\ °C$, the method is applied for the $CO_2$ separation (first two columns) and CH4 separation sections. According to the method described in the literature [41–43], S is chosen with reference to the tail column flow, namely, 12.06 and 14.34 mol/s for $CO_2$ and CH4, respectively. Hence, S is set at 45.6 and 19.8 ton/day, while f is set at 0.5, as reported in Table 4. The SOFC section cost is calculated as reported in the literature [39], including the auxiliary systems and inverter. Specifically, the stacks cost is calculated from the cell active area and the design temperature [44]. The relatively high capital cost for this configuration of the P2LSNG process is due to the DAC section, covering 47% of the initial investment. Biogas plants could be evaluated as a source of $CO_2$; in the DEC scenario, biomethane production from anaerobic digestion plants reaches 10.4 Gm3/y in 2040. Following the approach described in the literature [13], $CO_2$ availability could be estimated in 12.8 Mtonnes/y and be sufficient to cover the lot demand, calculated in 66,579 tonnes/y. Emission trading system (ETS) plants could represent another $CO_2$ source. This market reached 50.2 Mtonnes in 2019 [45]. Even if emissions reduction is expected in the 2040 DEC scenario for the ETS sector, $CO_2$ availability would be three orders of magnitude higher than the lot demand. As described in the literature [13], $CO_2$ sources near the P2LSNG systems could be exploited via existing technologies with a substantial plant cost reduction.

**Table 4.** P2LSNG process capital cost evaluation.

| Section | C (M€) | S | $C_0$ (M€) | $S_0$ | F |
|---|---|---|---|---|---|
| DAC | 21.32 | 22,897 tonnes/y | 549 [19] | $10^6$ tonnes/y [19] | 0.86 [20] |
| SOE | 12.4 | 21.5 MW | 23.17 [37] | 75 MW [37] | 0.5 [37] |
| Methanation | 1.69 | 21.5 MW | 1.08 [38] | 6 MW [38] | 0.35 [38] |
| TSA | 0.22 | 0.003469 kmol/s | 5.96 [36] | 0.294 kmol/s (purge gas) [36] | 0.74 [36] |
| Distillation ($CO_2$) | 5.34 | 45.6 tonnes/day | 33.94 [36] | 1839 tonnes/day [36] | 0.5 [36] |
| Distillation ($CH_4$) | 3.52 | 19.8 tonnes/day | 33.94 [36] | 1839 tonnes/day [36] | 0.5 [36] |
| SOFC | 0.39 [39] | 1.5 MW | - | - | - |
| Total | 44.88 | | | | |
| $CO_2$ Absorption [1] | 1.29 | 2.613 tonnes/h | 32.8 [36] | 327 tonnes/h [36] | 0.67 [36] |

[1] $CO_2$ capture in case of high intensity source available (e.g., power plant flue gas).

## 6. Conclusions

The concept of exploiting surplus energy in a high-RES scenario is investigated here at a national level through a recently developed P2LSNG process in order to produce a renewable fuel for HDV. It is highlighted, as the implementation of such a process does not require technological conversion at final users, while it allows for a higher HDV range because of the liquefied natural gas higher volumetric energy density with respect to CNG and hydrogen, both compressed and liquefied. Moreover, the produced LSNG can enable, after regasification, multiple uses in stationary applications, e.g., gas grid injection or power production, according to the low hydrogen content of the produced mixture (1.81%).

The hypotheses contained in the DEC 2040 scenario are used to define the available energy to feed the process and to satisfy the future LNG demand for HDV. This scenario does not include the locations of the critical electric transmission lines that are involved in curtailment operations; hence, the consequent areas for P2LSNG installation are not defined in the present study.

The process implementation at a national level was evaluated considering carbon dioxide feeding, both from local sources, and through the integration of a DAC section.

Nevertheless, for the hypothesis of full DAC technology development in 2040, it is estimated that the investment cost of the P2LSNG plant can be lowered by 44%, according to the data in Table 4, by removing the DAC section. This implies the availability of intensive $CO_2$ sources nearby to P2LSNG sites, e.g., biogas upgrade or ETS plants providing a $CO_2$ global availability at national level greater than the lot demand.

The plant lot operation reaches a power to chemical efficiency of 41.9% when a DAC section is integrated. Efficiency increases up to 51% when a local source of $CO_2$ is exploited to feed the methanation section, as described in the literature [13]. This would improve the produced fuel amount; moreover, the well-to-wheel analysis should be implemented to include $CO_2$ transport GHG emissions.

With the discussed DAC configuration, the LSNG production covers 4.2% of the national LNG expected demand according to the 2040 DEC scenario, while it satisfies only a residual share of national HDV traffic consumption (1.25%). GHG reduction with respect to diesel utilization reaches 88% for the covered demand.

Moreover, it is highlighted that P2LSNG plant lot operation can exploit 22.7% of the overgeneration, reducing its share in the DEC scenario. More accurate and profitable economical evaluations, including plant profitability and consequent lot sizing, could be possible when (i) local overgeneration is analyzed and (ii) P2LSNG are compared with other storage technologies to meet both short-term and seasonal grid flexibility requirements, improving RES integration.

In a previous work, the local analysis performed for the Italian Irpinia territory in the current scenario [13] resulted in a higher impact on transportation sector; the 8.5% of the HDV local traffic demand was estimated potentially covered by the production of a 21.5 MW P2LSNG plant exploiting a local $CO_2$ source. For the above, the $P_2$LNG technology could be considered to be suitable for local production under specific framework conditions (mainly local high wind production vs power request, beyond that availability of $CO_2$ sources), consequently to a proper mapping of the national territory.

**Author Contributions:** Conceptualization, L.B. and P.A.O.; methodology, L.B. and P.A.O.; software, M.P. and P.A.O.; validation, L.B., P.A.O. and M.P.; data curation, M.P. and P.A.O.; writing—original draft preparation, L.B. and P.A.O.; writing—review and editing, G.B., L.B. and P.A.O.; project administration, G.B. All authors have read and agreed to the published version of the manuscript.

**Funding:** This research received no external funding.

**Institutional Review Board Statement:** Not applicable.

**Informed Consent Statement:** Not applicable.

**Data Availability Statement:** Data sharing not applicable.

**Acknowledgments:** Terna S.p.A. is acknowledged for providing overgeneration data at 2040.

**Conflicts of Interest:** The authors declare no conflict of interest.

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
