# Peer review of "Liquefied Synthetic Natural Gas Produced through Renewable Energy Surplus: Impact Analysis on Vehicular Transportation by 2040 in Italy"

_2673-5628, doi:10.3390/gases1020007_

Round 1

Reviewer 1 Report

The study investigates a Power-to-Gas system, based on water electrolysis and CO2 methanation to produce hydro-methane, coupled with a low temperature gas upgrading section. The application of this power-to-liquefied synthetic natural gas systems is investigated at national level for Italy in 2040, assessing the number of plants to be installed to convert the curtailed energy. Synthetic fuel production avoided greenhouse gases emissions and the plant investment cost are evaluated.

The carried assessment is interesting and well in agreement with the Journal scope, even if the focus is only on a national/regional scale and generalization could be an improvement.

My minor comments on the manuscript, are:

  • in paragraph 2 (Methodology), the determination of the number of plants and of the maximum energy utilization is not clear and must be better explained.
  • Additional details could be provided on the DAC (CO2 production from pure air seems quite expensive, which is the state of the art of this technology).
  • Heat required for SOE operation is provided by the burner entirely or additional external heat is required? A comment could be added, this could affect the system efficiency.
  • In Table 1 the composition of the produced raw hydro-methane is indicated but what is the methane quantity in the final product at the end of the process?
  • The economic and environmental results are interesting and offer an insight on the P2G feasibility in the transport sector. Additional comments could be included for the case of possible stationary applications, mentioned the main similarities and differences.

Reviewer 2 Report

This paper is good and generally well written.  There is good detail given to the methodology. The individual units in the plant and capital cost evaluation are well explained. The conclusion section had just about the essential points. 

But,

The paper is lacking, in that it does not present a thorough scientific study.  There is no hypothesis, and therefore it is difficult to draw a strong conclusion. This is highlighted by the comment in the conclusions "P2LSNG technology, seams to be more suitable for local production......"

The results are not fully investigated. There is no presentation of how any of the independent variables will influence the results, and subsequently impact on the conclusion(s) drawn.  There is no table or graph with results from the simulation or their subsequent analysis.

While there is a mention of operational energy required, there is no mention of operational cost of the process. Subsequent to this there is no report on the final cost of the produced P2LSNG.  The paper would benefit substantially from this.  It would allow comparison with current and future diesel and/or alternative fuel costs.

The choice of SOE for the generation of hydrogen is not fully discussed or explained. It is well known from the literature the PEM electrolysers are more suited to the changing availability of renewable electricity and SOE are more suited to continuous operation. So it is unusual to see this choice here and needs a more detailed explanation. 

The choice of DAC as the source of CO2 is discussed, however its cost is 47% of the total cost. While biogas is mentioned as an alternative source of CO2, the impact of choosing DAC as opposed to biogas on the results and conclusions is not fully investigated or discussed.  It would be beneficial to discuss the potential change in plant efficiency when biogas is used in the methanator. Can the model be altered to include a different CO2 source? If so, it would be good to comment on that. It will potentially answer the previous comment and add additional information on the multi-applicability of the model.

There is no mention anywhere in the paper of the simulation software used in this research.

Other comments

There are some Typos including a number of Chemical formula which have not been given the correct subscript,   H2 & CO2. Even in the abstract

The term Hydromethane is used to describe SNG, Where as from my understanding Hydromethane is an unspecified Hydrogen Methane blend used in Transport. I suggest a more consistent use of terminology. 

Line 50: when compared to 

Line 58: When LNG is used as truck fuel

Line 71: Commercial vehicle diffusion is considered equal between.... (the use of diffusion is not appropriate in this context, consider rephrasing the sentence.

Line 82: Well to Wheel

Line 83: analysis presented here the P2LSNG

Line 85: the sentence beginning on this line is confusing and should be rephrased

Line 91:  "The latters were calculated ... "( I am not sure what is the meaning by the word latters here.  Suggest using the word results. )

Line 128: DELETE "improved to the hydromethane feeding"

Line 311:  "The concept of exploiting surplus energy in a high-RES scenario is investigated here at a national level through a recently developed P2LSNG process....

Line 319/320/321:  Consider rephrasing this sentence.

Line 330:  Consider rephrasing as it is not clear what is being said here.

Reviewer 3 Report

This manuscript treats liquefied synthetic natural gas produced through renewable energy surplus. Besides, the impact on vehicular transportation at 2014 in Italy is analyzed. Briefly, the manuscript is interesting and well written. For this reason, I recommend to publish it subjected to the following minor modifications: 

- Improve the introduction providing more related works. 

- Line 30, what is COP21? 

- Line 33, what is EC? 

- Explain more carefully how you have obtained Fig. 1. 

- Which units have the values shown in Table 1? Molar fraction? 

- Indicate which software you employed for Figs. 3 and 4. 

- Check the format of the references according to the rules of the journal.
